# Hepatitis E Prevalence in Vulnerable Populations in Goiânia, Central Brazil

**DOI:** 10.3390/v15102070

**Published:** 2023-10-10

**Authors:** Sheila Araújo Teles, Karlla Antonieta Amorim Caetano, Megmar Aparecida dos Santos Carneiro, Livia Melo Villar, Jeanne-Marie Stacciarini, Regina Maria Bringel Martins

**Affiliations:** 1Faculty of Nursing, Federal University of Goias, Goiania 74605-080, Brazil; karlla@ufg.br; 2Institute of Tropical Medicine and Public Health, Federal University of Goias, Goiania 74605-050, Brazil; megmar@ufg.br; 3Viral Hepatitis Laboratory, Oswaldo Cruz Institute, Rio de Janeiro 21045-900, Brazil; lvillar@ioc.fiocruz.br; 4College of Nursing, University of Florida, Gainesville, FL 32610-0197, USA; jeannems@ufl.edu

**Keywords:** hepatitis E virus, vulnerable populations, prevalence, Brazil

## Abstract

A transversal study was conducted among 472 vulnerable individuals (recyclable waste pickers, immigrants and refugees, homeless individuals, as well as lesbian, gay, bisexual, and transexual individuals) in Goiânia City, the capital of the State of Goiás, Brazil, to investigate the prevalence of hepatitis E virus (HEV) infection. A total of 459 (97.2%) serum samples were tested for anti-HEV IgG and IgM antibodies using fully automated chemiluminescence immunoassays (Liaison^®^ Murex Anti-HEV IgG and IgM assays, DiaSorin, Saluggia, Italy). Positive samples were tested for the presence of HEV RNA by a real-time polymerase chain reaction. A seroprevalence of 0.87% (95% confidence interval [CI]: 0.34–2.22) was found for anti-HEV IgG. Furthermore, anti-HEV IgM was detected in only one individual (0.22%; 95% CI: 0.04–1.22), who was also negative for HEV RNA. These findings revealed that HEV infection is infrequent in vulnerable individuals in Central Brazil, with low seroprevalence of past and recent HEV infections.

## 1. Introduction

Hepatitis E virus (HEV) infection is a significant public health concern in both developing and developed countries. The World Health Organization (WHO) estimates an annual global incidence of 20 million HEV infections, among which an estimated 3.3 million are symptomatic cases [1]. Although hepatitis E is usually a self-limiting acute infection, pregnant women and immunosuppressed individuals are at high risk of severe HEV-related morbidity. Pregnant women infected with HEV, particularly those in the second or third trimester, are at increased risk of fulminant hepatitis, fetal loss, and mortality. Cases of chronic hepatitis E have been reported in immunosuppressed individuals with subsequent cirrhosis, including mortality in some settings [2,3]. A vaccine to prevent hepatitis E (Hecolin^TM^) has been developed and is licensed in China and Pakistan but is not yet available elsewhere. Last year, however, the first-ever vaccine campaign was implemented in response to an HEV outbreak at Bentiu, South Sudan, Africa [1].

According to the current data of the International Committee on Taxonomy of Viruses (ICTV) [4], HEV belongs to the family *Hepeviridae*, subfamily *Orthohepevirinae*, genus *Paslahepevirus*, and species *Paslahepevirus balayani*. Although previously, HEV infection in humans was presumed to be limited to strains of *P. balayani*, recent studies have reported that the zoonotic transmission of *Rocahepevirus ratti* (a member of the *Orthohepevirinae* subfamily) is a new cause of hepatitis E in humans [5,6,7]), presumably resulting from contact with rodents, or their excrement.

HEV (*P. balayani*) includes eight different genotypes (1 to 8) and several subtypes, of which four main genotypes (1 to 4) can infect humans. Genotypes 1 and 2 are found exclusively in humans and are usually associated with waterborne infections in low- and middle-income countries with limited access to potable water, sanitation, hygiene, and health services. These genotypes are especially prevalent in Asia, Africa, and Mexico. In these areas, HEV infections occur both as outbreaks and as sporadic cases. Genotype 1 is associated more often with fulminant hepatitis and death in pregnant women. Otherwise, genotypes 3 and 4 infect humans and other animals, primarily domestic pigs. Human infections are mainly caused by the consumption of uncooked or undercooked animal meat (including animal liver, particularly pork). Therefore, hepatitis E is an emerging zoonotic infection in several European, North American, and South American countries, where genotype 3 can lead to chronic hepatitis and progress to cirrhosis in immunocompromised patients [8]. In Brazil, genotype 3 is the only genotype that shows a high homology between human and swine isolates, thus providing evidence of the zoonotic transmission of this virus and its importance in the One Health Initiative founded by the WHO [9].

In South America, epidemiological studies on HEV infection (serological and molecular screening) among blood donors have reported a prevalence of 3.5% (95% CI: 2.1–5.4) in Argentina [10] and 10.0% (95% CI: 7.2–13.3) in Uruguay [11]. Based on the results of their systematic review and meta-analysis of hepatitis E seroprevalence in adults in Brazil, Tengan et al. [12] estimated an overall seroprevalence of 6.0% (95% CI: 5.0–7.0); in subgroup analyses, the authors found that the prevalence of anti-HEV in blood donors was 7.0% (95% CI: 5.0–8.0), while in the general population, this was 3.0% (95% CI: 2.0–4.0). However, the seroprevalence rates in the general population and blood donors in Brazil vary widely [9,13], with the lowest (0.44%) observed in blood donors in Manaus, Amazonas (North Region) [14] and the highest (65.50%) in the general population of Passo Fundo, Rio Grande do Sul (South Region) [15]. As reported in another study [12], differences in sensitivity and specificity between the anti-HEV immunoassays could influence HEV seroprevalence data in epidemiological studies. Thus, more studies using HEV serological assays with good performance and molecular assays are needed to understand the epidemiology of HEV infections in Brazil, a continental country. To address this, this study aimed to investigate the prevalence of HEV infection in socially vulnerable populations in Central Brazil.

## 2. Materials and Methods

### 2.1. Study Population

During the first wave of the COVID-19 pandemic, the Federal University of Goias/Universidade Federal de Goiás developed a community project for screening SARS-CoV-2, free of cost, for health and safety professionals. To support this, a strict biosafety infrastructure was created on the university’s premises. Accordingly, a cross-sectional study was carried out between July and October 2020 to investigate SARS-CoV-2 infection in socially vulnerable populations in Goiânia (the capital of the State of Goiás), Central Brazil, as previously described by Santos et al. [16]. Additionally, individuals over the age of 18 were invited to participate in the study. Those who agreed provided written informed consent. The participants were interviewed, and a blood sample was collected to detect HEV markers. This project was approved by the Research Ethics Committee (process no. 4145237).

### 2.2. Laboratory Tests

Of the 472 participants, in 13 participants their serum samples were insufficient for analysis. As a result, a total of 459 (97.2%) serum samples were tested using fully automated chemiluminescence immunoassays for the detection of anti-HEV immunoglobulin G (IgG) and immunoglobulin M (IgM) (Liaison^®^ Murex Anti-HEV IgG and IgM assays, DiaSorin, Saluggia, Italy), which, according to the manufacturer, have 97.92% and 98.44% sensitivity and 99.22% and 97.93% specificity for anti-HEV IgG and IgM assays, respectively. Both assays were coated with genotype 1 and 3 ORF2 antigens, and 100 µL of serum samples were used in each assay. Quantitative Liaison^®^ Murex HEV IgG measures the intensity of the luminescence, which is proportional to the concentration of HEV IgG in the serum sample. Test values are automatically calculated by the instrument and expressed as IU/mL IgG values below the threshold value (0.3 IU/mL) were considered to be negative. Regarding the qualitative Liaison^®^ Murex HEV IgM assay, values obtained with this assay are automatically calculated by the instrument as an index. Samples with an IgM index value below the threshold value (set at 1.00) were considered to be negative.

Anti-HEV IgG/IgM-positive samples were tested for the presence of hepatitis E virus ribonucleic acid (HEV RNA), which was extracted from 140 μL of serum using a QiaAmp viral RNA kit (Qiagen, Hilden, Germany), according to the manufacturer’s instructions. RNA reverse transcription (RT) was conducted using Superscript III (Invitrogen, Carlsbad, CA, USA) at 50 °C [17]. Real-time polymerase chain reaction (PCR) was performed for the detection (limit of four copies of HEV RNA per reaction) and quantification of HEV RNA using TaqMan real-time PCR technology [18]. HEV plasmid DNA (constructed from a Brazilian swine HEV strain previously characterized) was quantified using the Nanodrop ND-1000 instrument, according to the manufacturer’s instructions (Wilmington, DE, USA). Standard curves were generated using 10^0^ to 10^9^ copies of plasmid DNA. The genome equivalent titers of HEV were determined based on the standard curve [17].

### 2.3. Data Analysis

Data were analyzed using the IBM Statistical Package for Social Science (SPSS) (IBM SPSS Statistics for Windows, Version 28.0). The relative and absolute frequencies and the tendency central measures (median and interquartile range) were calculated for nominal and continuous variables, respectively. Prevalence was estimated using a confidence interval of 95% (95% CI).

## 3. Results

A total of 472 individuals participated in the study. As shown in Table 1, the median age of the study population was 34 years (interquartile range [IQR], 22 years). The majority were male (57.8%), self-declared mixed color/race (50.2%), and single (60.9%), with reportedly less than a secondary education level (52.6%; 11 or fewer years of education) and a median monthly income of USD 223 (IQR: 212). They were recyclable waste pickers (31.1%), immigrants/refugees (25%), homeless who reported sleeping in a public shelter (23.3%), lesbian, gay, bisexual, transexual (LGBT) individuals (16.7%), and other (sex workers, individuals who use illicit drugs, and individuals living with HIV) (3.8%).

Among the 459 participants tested for anti-HEV IgM and IgG markers, four individuals were anti-HEV IgG-positive (0.87%; 95% CI: 0.34–2.22), while one was also positive for anti-HEV IgM (0.2%; 95% CI: 0.04–1.22) but negative for HEV RNA.

The four anti-HEV IgG-positive individuals were aged between 29 and 63 years, among which two were males, and two were females, with a monthly income of USD 111. All but one self-declared as black or mixed color/race and reported having incomplete primary education (≤8 years of schooling). Furthermore, two unemployed individuals were sheltered homeless, while two were recyclable waste pickers, one of whom (ID-378) was also positive for anti-HEV IgM (Table 2).

## 4. Discussion

Viral hepatitis remains a major public health problem, significantly impacting morbidity and mortality on a global scale. A WHO study found that an estimated 4.5 million premature deaths could be prevented in low- and middle-income countries by 2030 through education campaigns, vaccination, testing, and treatment. At present, the World Hepatitis Alliance and WHO are promoting strategies for the elimination of viral hepatitis as a public health problem, with the aim of reducing new hepatitis infections by 90% and deaths from viral hepatitis by 65% between 2016 and 2030. To achieve the goals established by the WHO in its global hepatitis strategy, it is essential to bring hepatitis care close to communities [19], especially vulnerable subpopulations. Additionally, the clinical recognition and accurate diagnosis of HEV infection are essential for the management of hepatitis E [3,8]. Additionally, HEV vaccination and other control measures are needed to limit future hepatitis E outbreaks [1,20].

Among the participants enrolled in this study, a prevalence of 0.87% (95% CI: 0.34–2.22) for anti-HEV was observed. This rate is within the large range of hepatitis E seroprevalence rates observed in Brazil. In the North Region, Kiesslich et al. [14] found a prevalence of 0% among 100 pregnant women, 0.44% among 227 blood donors, and 0.51% among 192 hemodialysis patients in Amazonas, while Villa et al. [21] observed seroprevalence rates of 0.19% and 0% among 506 and 175 indigenous and non-indigenous individuals, respectively, in Tocantins. Similarly, in the Northeast Region, Parana et al. [22] reported a prevalence of 0% among 392 hemodialysis patients in Salvador, Bahia, while in the Southeast Region, Trinta et al. [23] found a seroprevalence of 0% among individuals living in the urban area of Rio de Janeiro. Nevertheless, Zorzetto et al. [14] reported the highest anti-HEV prevalence in Passo Fundo (65.5%; 665/1000), followed by Caxias do Sul (57.4%; 574/1000) and Santa Maria (55.4%; 554/1000), Rio Grande do Sul (South Region). Also, in relation to the variability of HEV prevalence in Brazil, in their systematic review and meta-analysis of 20 selected studies of hepatitis E seroprevalence in adults in Brazil, Tengan et al. [12] observed a range from 0.0% (95% CI: 0.0–3.0) to 10.0% (95% CI: 7.0–15.0). This variation is most likely due to the different characteristics of the population and the sensitivity of diagnostic tests used for the detection of anti-HEV antibodies. Despite these differences, these findings highlight the variability of HEV seroprevalence depending on the geographical area and population studied in Brazil [9,13].

Although infectious diseases primarily affect socially and economically vulnerable individuals [24], the prevalence of anti-HEV found among these individuals in this study was low. Consistent with this finding, relatively low HEV seroprevalence rates were observed in other groups of highly vulnerable individuals, including individuals exposed to unfavorable socio-economic and environmental factors in Central Brazil, such as recyclable waste pickers (5.1%; 95% CI: 3.4–7.6) [25] and rural residents of settlement projects (3.9%; 95% CI: 2.8–5.4) [26]. In contrast, high hepatitis A virus (HAV) seroprevalence rates have been previously reported among these vulnerable groups (99.5% and 85.9%, respectively) [26,27], suggesting that, in contrast to hepatitis A, HEV infection in Brazil does not seem to be associated with low income and suboptimal hygiene and sanitation conditions. Moreover, although these findings are consistent with those reported by de Oliveira et al. [9] in a recent review that focused on HEV seroprevalence data from distinct Brazilian populations, more comparative data on HEV epidemiology of these enterically transmitted viruses are needed.

Of the four anti-HEV IgG-seropositive individuals, two were homeless and reported sleeping in a public shelter. The rare studies on HEV infection prevalence among homeless individuals have reported a slightly higher rate, such as in Tehran, Iran, than that found in the general population (24.3% vs. 9.3%) [28], as well as in Marseille, France (11.6%) [29], suggesting that homelessness increases the risk of exposure to HEV. However, to the best of our knowledge, there are currently no studies on the seroprevalence of hepatitis E in homeless individuals in Brazil. This study is the first to report HEV seroprevalence in a group of Brazilian sheltered homeless individuals (2/110; 1.8%). Two other anti-HEV IgG-seropositive individuals were recyclable waste pickers, one of whom was also anti-HEV IgM-positive but negative for HEV RNA. As observed in other studies [2,25], the absence of HEV RNA can be explained by the short period of viraemia in cases of self-limited infection, which seems to be the profile of this individual. As previously reported, recyclable waste pickers in Brazil collect, separate, classify, and sell all types of recyclable waste materials. Most of these workers have reported consuming non-filtered water and food sources from the garbage, as well as coming into contact with human and animal feces and using gloves and other personal protective equipment inconsistently [25,27]. Therefore, these potential risk behaviors for HEV transmission may have contributed to the past and recent cases of HEV infection in this study.

In this study, it is noteworthy that the recyclable waste picker who was positive for anti-HEV IgG and IgM had been exposed to HAV (antibodies to HAV, total anti-HAV positive), hepatitis B virus (antibodies to the total hepatitis B core antigen, anti-HBc positive), and hepatitis C virus (antibodies to HCV, anti-HCV positive). In addition, he also tested positive for COVID-19 (RNA-SARS-CoV-2) (data not shown). After post-test counseling, he was referred for clinical evaluation. Unfortunately, this scenario of exposure to several infectious diseases is the result of the combination of multiple risk behaviors in the context of the life and work of recyclable waste pickers [25,27].

This study has some limitations that should be considered. First, the convenience sampling used may have compromised the external validity of the results. Moreover, the challenges associated with accessing this population and the health restrictions imposed by the COVID-19 pandemic during this study should be noted. Second, the collection of data on contact with animals, as well as eating habits and other risk behaviors, does not occur in a systematic way, making it difficult to analyze the results. Third, an in-house real-time PCR was used to detect HEV RNA in the serum samples of anti-HEV IgG/IgM-positive individuals, which was additionally not examined in their corresponding stool samples. Furthermore, it was not possible to investigate the presence of HEV RNA in anti-HEV-negative participants. As reported previously [30], the detection of HEV RNA or active infection without serological evidence of infection could be neglected due to the fact that viral RNA appears ahead of IgG and IgM antibodies. Nonetheless, despite these limitations, given the fact that very little data exists on the epidemiology of HEV infection in Central Brazil, this study provides important findings on the prevalence of HEV in a socially vulnerable population.

As there is no gold standard for the measurement of anti-HEV, its prevalence can be under- or overestimated depending on the sensitivity and specificity of the assay used [12]. In this study, a relatively low prevalence of anti-HEV IgG and IgM was observed using Liaison^®^ assays for the first time in Brazil. To the best of our knowledge, only one study has previously evaluated these new assays to date. Recently, Abravanel et al. [31] reported that the Liaison^®^ Murex assays are very sensitive and specific in comparison to the Wantai HEV assays, observing a rate of sensitivity of 100% for IgM and IgG assays with acute-phase samples from immunocompetent patients, although less for IgM (93.8%) and IgG (84.3%) assays obtained from immunocompromised patients. In addition, a sensitivity of 94.6% was observed for anti-HEV IgG and 57.4% for anti-HEV IgM using post-viremic samples from immunocompetent patients, which in turn was 93.5% and 71% for immunocompromised patients. The corresponding specificity was very high (>99%). Furthermore, the authors confirmed the limit of detection stated for the IgG assay (0.3 U/mL), demonstrating that Liaison^®^ IgG assay is suitable for use in seroprevalence studies. Therefore, further studies on the HEV seroprevalence in Brazil using these new assays are needed, especially in other Brazilian regions with heterogeneous epidemiological profiles.

In conclusion, in terms of the epidemiology of hepatitis E in Central Brazil, this study revealed a low prevalence of past and recent HEV infections in the socially vulnerable population studied.

## Figures and Tables

**Table 1 viruses-15-02070-t001:** Sociodemographic characteristics of 472 vulnerable individuals in Central Brazil.

Characteristics	N = 472	(%)
**Age** (median; IQR *)	34 (22)	
**Monthly income** (median; IQR)	223 (212)	
**Sex**		
Female	199	42.2
Male	273	57.8
**Education**		
Incomplete primary education	160	33.9
Primary education	40	8.5
Incomplete secondary education	48	10.2
Secondary education	155	32.8
>Secondary education	69	14.6
**Marital status**		
Married	138	29.8
Single	282	60.9
Widow/divorced	43	8.7
**Color/race**		
White	95	20.1
Black	109	23.1
Mixed	237	50.2
Asian/indigenous	31	6.6
**Population subgroup**		
Recyclable waste pickers	147	31.1
Immigrants/Refugees	118	25.0
Homeless	110	23.3
LGBT **	79	16.7
Other	18	3.8

* IQR: Interquartile range; ** LGBT: lesbian, gay, bisexual, transexual.

**Table 2 viruses-15-02070-t002:** Characteristics of anti-HEV-positive vulnerable individuals in Central Brazil.

Variable	ID-373	ID-378	ID-493	ID-494
**Age**	63	57	49	29
**Sex**	Female	Male	Male	Female
**Color**	Black	Mixed	Black	White
**Marital status**	Single	Widow	Married	Married
**Education ***	IPL	IPL	SE	IPL
**Monthly income (USD)**	111	111	111	111
**Employment ****	RWP	RWP	Unemployed	Unemployed
**Anti-HEV IgM**	No	Yes	No	No

* IFL: Incomplete primary education; SE: Secondary education. ** RWP: recyclable waste picker.

## Data Availability

The data supporting the findings of this study are available from the corresponding author upon reasonable request.

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
