# Peer review of "Hepatitis E Prevalence in Vulnerable Populations in Goiânia, Central Brazil"

_viruses, 2023, doi:10.3390/v15102070_

Round 1

Reviewer 1 Report

The manuscript “Hepatitis E seroprevalence in vulnerable population in Goiânia, Central Brazil” provides data on the seroprevalence of Hepatitis E Virus in humans in one town in Central Brazil. I think that the study is limited and does not provide sufficient data to compare and analyze the HEV seroprevalence in Brazil.

A careful revision of the English language is necessary to correct grammatical and punctuation errors and improve the level of English.

More detailed information on the prevalence of HEV (serological and molecular studies) in Brazil and neighboring countries needs to be included in the introduction.

L27- Please, rewrite the sentence, it is too complicated to understand.

L31 – Please, rewrite the sentence, the vaccine is licensed in China and Pakistan,  

 A vaccine to prevent hepatitis E (HecolinTM) has been developed and is licensed

Please include more information about HEV taxonomy, Paslahepevirus balayani, and Rocahepevirus ratti and their strains that cause acute hepatitis in humans.

L58-71 - This paragraph is confusing. Please, better organize the information and rephrase it in a clearer way

L73 - Please, explain the selection criteria for testing of 459 samples, not all 472 samples.

L76 – It is one assay for simultaneous testing of IgG and IgM, or both assays? Please give more information

L80- Please, clarify which positive samples you checked, all or only IgM?

L183 – It is not clear if the sample which is positive for IgM is positive for IgG. Please, clarify this.

Discussion L130-141 Please, rewrite the paragraph and include the date from the systematic review of Tengan et al. Seroprevalence of hepatitis E in adults in Brazil: a systematic review and meta-analysis.

A careful revision of the English language is necessary to correct grammatical and punctuation errors and improve the level of English.

Author Response

Review 1

The manuscript “Hepatitis E seroprevalence in vulnerable population in Goiânia, Central Brazil” provides data on the seroprevalence of Hepatitis E Virus in humans in one town in Central Brazil. I think that the study is limited and does not provide sufficient data to compare and analyze the HEV seroprevalence in Brazil.

A careful revision of the English language is necessary to correct grammatical and punctuation errors and improve the level of English. More detailed information on the prevalence of HEV (serological and molecular studies) in Brazil and neighboring countries needs to be included in the introduction.

Response: Thank you very much for your comments. We have reviewed the language and indeed found many opportunities for improvement. Taking your comment into consideration, further information on the prevalence of HEV in Brazil and neighboring countries has been included in the Introduction section of the revised manuscript.

L27- Please, rewrite the sentence, it is too complicated to understand.

Response: This sentence has been rewritten.

L31 – Please, rewrite the sentence, the vaccine is licensed in China and Pakistan.

A vaccine to prevent hepatitis E (HecolinTM) has been developed and is licensed

Response: This sentence has been rewritten.

Please include more information about HEV taxonomy, Paslahepevirus balayani, and Rocahepevirus ratti and their strains that cause acute hepatitis in humans.

Response: Further information on the taxonomy of HEV has been included in the Introduction section.

L58-71 - This paragraph is confusing. Please, better organize the information and rephrase it in a clearer way.

Response: The paragraph has been rewritten.

L73 - Please, explain the selection criteria for testing of 459 samples, not all 472 samples.

Response: Unfortunately, for 13 of the participants, the volume of their serum samples was insufficient for analysis.

L76 – It is one assay for simultaneous testing of IgG and IgM, or both assays? Please give more information.

Response: Liaison® Murex Anti-HEV IgG and IgM assays (DiaSorin, Saluggia, Italy). Taking your comment into consideration, we have included information regarding these assays to the revised manuscript in the Materials and Methods section (“Laboratory tests” section).

L80- Please, clarify which positive samples you checked, all or only IgM?

Response: All anti-HEV positive samples (IgM and IgG).

L183 – It is not clear if the sample which is positive for IgM is positive for IgG. Please, clarify this.

Response: OK! The sample (ID-378) was positive for IgG and IgM. This has been clarified in the Results and Discussion sections of the revised manuscript.

Discussion L130-141 Please, rewrite the paragraph and include the date from the systematic review of Tengan et al. Seroprevalence of hepatitis E in adults in Brazil: a systematic review and meta-analysis.

Response: The paragraph has been rewritten, and the data from the systematic review of Tengan et al. (“Seroprevalence of hepatitis E in adults in Brazil: a systematic review and meta-analysis”) has been included in the revised manuscript.

A careful revision of the English language is necessary to correct grammatical and punctuation errors and improve the level of English.

Response: The English language has been carefully revised, and many opportunities have been found for improvement.

Reviewer 2 Report

Title Hepatitis E seroprevalence in vulnerable population in Goiânia, Central Brazil

Authors have performed a transversal study on 472 vulnerable individuals in Goiânia City in Brazil to investigate the prevalence of HEV. This is an interesting set of valuable data on a virus that has its studies still on infancy when considering particularly exposed or vulnerable population. Besides testing sera samples for IgG and IgM using a clia, authors have provided a comprehensive text on the subject. In my opinion the manuscript will be well received by the scientific community. The manuscript is well written. I have some (very) minor questions below and advise accepting with minor revisions

Please explain how quantification and detection limits were performed. Did you use a stardard (RNA or DNA)?

Interesting to see such a low seroprevalence. There is an interesting (recent) review “ Systematic Review of Hepatitis E Virus in Brazil: A One-Health Approach of the Human-Animal-Environment Triad” that deserves to be mentioned for comparison purposes on HEV seropositivity.

Would suggest a slight change of the title to include the fact that you did also molecular screening

Author Response

Review 2

Authors have performed a transversal study on 472 vulnerable individuals in Goiânia City in Brazil to investigate the prevalence of HEV. This is an interesting set of valuable data on a virus that has its studies still on infancy when considering particularly exposed or vulnerable population. Besides testing sera samples for IgG and IgM using a clia, authors have provided a comprehensive text on the subject. In my opinion the manuscript will be well received by the scientific community. The manuscript is well written. I have some (very) minor questions below and advise accepting with minor revisions.

Response: Thank you for your comments.

Please explain how quantification and detection limits were performed. Did you use a standard (RNA or DNA)?

Response: HEV plasmid DNA (constructed from a Brazilian swine HEV strain previously characterized) was quantified with the Nanodrop ND-1000 instrument, according to manufacturer’s instructions (Wilmington, DE). Standard curves were generated using 100 to 109 copies of plasmid DNA. The genome equivalent titers of HEV were determined based on the standard curve (dos Santos et al., 2011).

Interesting to see such a low seroprevalence. There is an interesting (recent) review “Systematic Review of Hepatitis E Virus in Brazil: A One-Health Approach of the Human-Animal Environment Triad” that deserves to be mentioned for comparison purposes on HEV seropositivity.

Response: Data from the systematic review conducted by Moraes et al. (“Systematic Review of Hepatitis E Virus in Brazil: A One-Health Approach of the Human-Animal Environment Triad”) have been included in the Discussion section of the revised manuscript.

Would suggest a slight change of the title to include the fact that you did also molecular screening.

Response: “Hepatitis E prevalence in vulnerable population in Goiânia, Central Brazil.”

Reviewer 3 Report

Overview and general recommendation: The simple present study describes the HEV seroprevalence in a total of 472 vulnerable individuals in a central Brazilian city. As a result, the HEV seroprevalence is low with anti-HEV IgG (0.87%) and IgM (0.22%), and no HEV RNA was detected. Altogether, the authors conclude that HEV infection is infrequent in vulnerable individuals in Central Brazil. Overall, the study is helpful in understanding the HEV infection in Brazil; however, too limited data would depreciate its significance and soundness. My comments are below.

Specific points:

1.     Abstract: “vulnerable population” is a huge concept. The authors should emphasize which specific groups of individuals are involved and analyzed in this study.

2.     Line 18: change “0.2%” to “0.22%” according to “0.87%” outlined earlier.

3.     Since the authors have compared and discussed the variability of HEV seroprevalence in different geographical areas of Brazil, a map would be beneficial for the readers.

4.     Section 2.2, line 73: curiously, why were 459 of 472 serum samples tested? Notably, in Table 1, the number is still 472; which 13 samples are absent for the HEV seroprevalence test?

5.     Line 80: if the reviewer understands correctly, four anti-HEV IgG-positive samples were tested for HEV RNA. Is there any possibility some HEV RNA-positive individuals are neglected due to the fact that viral RNA appears ahead of IgG and IgM antibodies (please refer to PMID: 27527210)? In this scenario, it should be cautious to conclude the absence of active HEV infection (line 21).

6.     Lines 149 to 150: this sentence seems overstated to me. It is difficult to draw the conclusion that HEV infection in Brazil was not associated with low income and suboptimal and sanitation conditions based on one or two studies. Importantly, no comparative data is available on HEV infection in apparently healthy individuals from this study.

Minor editing of English language required

Author Response

Review 3

Overview and general recommendation: The simple present study describes the HEV seroprevalence in a total of 472 vulnerable individuals in a central Brazilian city. As a result, the HEV seroprevalence is low with anti-HEV IgG (0.87%) and IgM (0.22%), and no HEV RNA was detected. Altogether, the authors conclude that HEV infection is infrequent in vulnerable individuals in Central Brazil. Overall, the study is helpful in understanding the HEV infection in Brazil; however, too limited data would depreciate its significance and soundness. My comments are below.

Response: Thank you very much for your comments, which are addressed below.

Specific points:

  1. Abstract: “vulnerable population” is a huge concept. The authors should emphasize which specific groups of individuals are involved and analyzed in this study.

Response: We have specified the vulnerable subpopulations evaluated in our study in the Abstract of the revised manuscript.

  1. Line 18: change “0.2%” to “0.22%” according to “0.87%” outlined earlier.

Response: OK! It was changed.

  1. Since the authors have compared and discussed the variability of HEV seroprevalence in different geographical areas of Brazil, a map would be beneficial for the readers.

Response: Due to the limitations on the number of Figures and Tables for “Brief reports,” we have chosen to include two tables in our manuscript.

  1. Section 2.2, line 73: curiously, why were 459 of 472 serum samples tested? Notably, in Table 1, the number is still 472; which 13 samples are absent for the HEV seroprevalence test?

Response: 472 people agreed to participate in the study, from which blood samples were collected. However, among these, the volume of the serum sample of 13 individuals was not sufficient for analysis.

  1. Line 80: if the reviewer understands correctly, four anti-HEV IgG-positive samples were tested for HEV RNA. Is there any possibility some HEV RNA positive individuals are neglected due to the fact that viral RNA appears ahead of IgG and IgM antibodies (please refer to PMID: 27527210). In this scenario, it should be cautious to conclude the absence of active HEV infection (line 21).

Response: We agree with the reviewer’s comment, and the corresponding sentence in the Abstract was changed. Also, considering the given information above, this has been added as a limitation of our study.

  1. Lines 149 to 150: this sentence seems overstated to me. It is difficult to draw the conclusion that HEV infection in Brazil was not associated with low income and suboptimal and sanitation conditions based on one or two studies. Importantly, no comparative data is available on HEV infection in apparently healthy individuals from this study.

Response: We agree with your comment. The sentence has been rewritten.

Minor editing of English language required

Response: Thank you. The English language has been carefully revised, and several areas warranting improvement have been identified.

Round 2

Reviewer 1 Report

It seems that the quality of the presentation of the manuscript has improved.

L39 Start the description of the classification of the virus from the family, it is too detailed.

L145 The information in the sentence is repeated, please include more information or delete the sentence. 

It seems that the quality of the presentation of the manuscript has improved.

Author Response

Review 1

Comments and Suggestions for Authors

It seems that the quality of the presentation of the manuscript has improved.

Response: Thank you very much for your comments.

L39 Start the description of the classification of the virus from the family, it is too detailed.

Response: We have started the description of the classification of the virus from the family.

L145 The information in the sentence is repeated, please include more information or delete the sentence. 

 Response: The sentence has been rewritten.

Comments on the Quality of English Language

It seems that the quality of the presentation of the manuscript has improved.

Response: Thank you for your carefully revision of our manuscript.

Reviewer 3 Report

The authors have adequately addressed my questions and queries, I have no more comments.

Author Response

Review 3

Comments and Suggestions for Authors

The authors have adequately addressed my questions and queries, I have no more comments.

Response: Thank you for your carefully revision of our manuscript.
